# Trajectory VAE for Multi-Modal Imitation

## Abstract

We address the problem of imitating multi-modal expert demonstrations in sequential decision making problems. In many practical applications, for example video games, behavioural demonstrations are readily available that contain multi-modal structure not captured by typical existing imitation learning approaches. For example, differences in the observed players' behaviours may be representative of different underlying playstyles.

In this paper, we use a generative model to capture different emergent playstyles in an unsupervised manner, enabling the imitation of a diverse range of distinct behaviours. We utilise a variational autoencoder to learn an embedding of the different types of expert demonstrations on the trajectory level, and jointly learn a latent representation with a policy. In experiments on a range of 2D continuous control problems representative of Minecraft environments, we empirically demonstrate that our model can capture a multi-modal structured latent space from the demonstrated behavioural trajectories.

## 1 Introduction

Imitation learning has become successful in a wide range of sequential decision making problems, in which the goal is to mimic expert behaviour given demonstrations (Ziebart et al., 2008; Wang et al., 2017; Li et al., 2017; D'Este et al., 2003). Compared with reinforcement learning, imitation learning does not require access to a reward function – a key advantage in domains where rewards are not naturally or easily obtained. Instead, the agent learns a behavioural policy implicitly through demonstrated trajectories.

Expert demonstrations are typically assumed to be provided by a human demonstrator and generally can vary from person to person, e.g., according to their personality, experience and skill at the task. Therefore, when capturing demonstrations from multiple humans, observed behaviours may be distinctly different due to multi-modal structure caused by differences between demonstrators. Variations like these, which are very common in video games where players often cluster into distinct play styles, are typically not modelled explicitly as the structure of these differences is not known a priori but instead emerge over time as part of the changing meta-game.

In this paper, we propose *Trajectory Variational Autoencoder (T-VAE)* a deep generative model that learns a structured representation of the latent features of human demonstrations that result in diverse behaviour, enabling the imitation of different types of emergent behaviour. In particular, we use a *Variational Autoencoder (VAE)* to maximise the *Evidence Lower Bound (ELBO)* of the log likelihood of the expert demonstrations on the trajectory level where the policy is directly learned from optimising the ELBO. Not only can our model reconstruct expert demonstrations, but we empirically demonstrate it learns a meaningful latent representation of distinct emergent variances in the observed trajectories.

## 2 Related Work

Popular imitation learning methods include *behavior cloning (BC)* (Pomerleau, 1991), which is a supervised learning method that learns a policy from expert demonstration of state-action pairs. However, this approach assumes independent observations which is not the case for sequential

decision making problems, as future observations depend on previous actions. It has been shown that BC cannot generalise well to unseen observations (Ross & Bagnell, 2010). Ross et al. (2011) proposed a new iterative algorithm, which trains a stationary deterministic policy with no regret learning in an online setting to overcome this issue. Torabi et al. (2018) also improve behaviour cloning with a two-phase approach where the agent first learns an inverse dynamics model via interacting with the environment in a self-supervised fashion, and then use the model to infer missing actions given expert demonstrations. An alternative approach is *Apprenticeship Learning (AC)* (Abbeel & Ng, 2004), which uses inverse reinforcement learning to infer a reward function from expert trajectories. However, it suffers from expensive computation due to the requirement of repeatedly performing reinforcement learning from tabula-rasa to convergence. Whilst each of these methods has had successful applications, none are able to capture multi-modal structure in the demonstration data representative of underlying emergent differences in playstyle.

More recently, the learning of a latent space for imitation learning has been studied in the literature. *Generative Adversarial Imitation Learning (GAIL)* (Ho & Ermon, 2016) learns a latent space of demonstrations with a *Generative Adverserial Network (GAN)* (Goodfellow et al., 2014) like approach which is inherently mode-seeking and does not explicitly model multi-modal structure in the demonstrations. This limitation was addressed by (Li et al., 2017), who built on the GAIL framework to infer a latent structure of expert demonstrations enabling imitation of diverse behaviours. Similarly, (Wang et al., 2017) combined a VAE with a GAN architecture to imitate diverse behaviours. However, these methods require interacting with the environment and rollouts of the policy whilst learning. For comparison we note our method does not need access to the environment simulator during training and is computationally cheaper, as the policy is learned simply by gradient descent using a fixed dataset of trajectories. Additionally, whilst the aim in GAIL is to keep the agent behaviour close to the expert's state distribution, our model can serve as an alternative approach to capturing state sequence structure.

In work more closely related to our approach, (Co-Reyes et al., 2018) have also proposed a *Variational Auto encoder (VAE)* (Kingma & Welling, 2013) that embeds the expert demonstration on the trajectory level which showed promising results. However their approach only encodes the trajectories of the states whereas ours encodes both the state and action trajectories, which also allows us to learn the policy directly from the probabilistic model rather than adding a penalty term to the ELBO. Rabinowitz et al. (2018) also learns an interpretable representation of the latent space in a hierarchical way, but their focus is more on representing the mental states of other agents and is different from our goal of imitating diverse emergent behaviours.

## 3 METHODS

### 3.1 PRELIMINARIES

Let the tuple $(\mathcal{S}, \mathcal{A}, P, r, I)$ denote the infinite-horizon *Markov Decision Processes (MDP)* with: $\mathcal{S}$ the state space, $\mathcal{A}$ the action space, $P$ the transition probability distribution, $r$ the reward function and $I$ the distribution of the initial state $s_0$. Let $\pi_E : \mathcal{S} \times \mathcal{A} \to [0, 1]$ denote the expert policy which we do not know, under which expert trajectories $\tau$ of states and actions are generated from, *i.e.*, $s_0 \sim I, a_t \sim \pi_E(a_t|s_t), s_{t+1} \sim P(s_{t+1}|a_t, s_t)$. The goal of imitation learning is to learn a policy $\pi$ that best explains the trajectories without knowledge of the reward signal $r$.

### 3.2 TRAJECTORY VAE (T-VAE)

Given $N$ demonstrated trajectories $\{\tau^{(i)}\}_{i=1}^{N}$ of states and actions, where each $\tau^{(i)} = \{(s_t^{(i)}, a_t^{(i)})\}_{t=1}^{T_i}$, where $T_i$ is the length of trajectory $\tau^{(i)}$. The marginal likelihood of the set of trajectories is composed of a sum over the marginal likelihoods of each individual trajectory $\log p_\theta(\tau^{(1)}, \cdots, \tau^{(N)}) = \sum_{i=1}^{N} \log p_\theta(\tau^{(i)})$. We use a latent variable model and assume the prior distribution of latent $z$ is standard Gaussian. Rather than a VAE which is applied on the data point level, we use a VAE on the trajectory level (which consists of a time sequence of data points), as shown in Figure 1a. We call our model *Trajectory VAE (T-VAE)*.

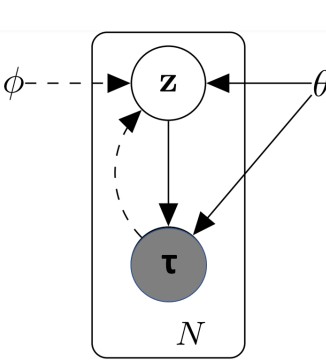

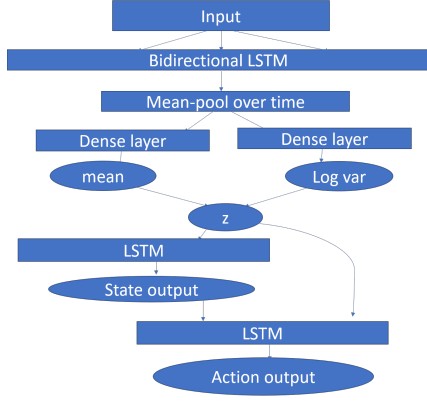

(a) Graphical model of T-VAE    (b) Computation graph of T-VAE

### 3.2.1 ENCODER NETWORK

We encode whole trajectories into the latent space in order to embed useful features of different behaviours and extract distinguishing features which differ from trajectory to trajectory. Note that the latent $z$ is therefore a single variable rather than a sequence that depends on $t$. In order to utilise all information, we encode both the states and actions, i.e., $q_\phi(z|\tau^{(i)}) = q_\phi(z|\{(s_t^{(i)}, a_t^{(i)})\}_{t=1}^{T_i})$. We assume that the approximate posterior $q_\phi(z|\tau^{(i)})$ has a Gaussian distribution, whose mean and log variance parameters are constructed as follows (and illustrated in the top half of Figure 1b): 1) concatenate states with actions at each time step $[s_t, a_t]$; 2) feed the sequence into a bidirectional LSTM; 3) mean-pool over the outputs along the time horizon ($T^{(i)}$); 4) pass through two separate fully connected layers.

### 3.2.2 DECODER NETWORK

The decoder $p(\tau^{(i)}|z)$ can be decomposed as

$$p_\theta(\tau^{(i)}|z) = p_{\theta_{SD}}(\{s_t^{(i)}\}_{t=1}^{T}|z)p_{\theta_{PD}}(\{a_t^{(i)}\}_{t=1}^{T}|z, \{s_t^{(i)}\}_{t=1}^{T}) \qquad (1)$$

where we call $p_{\theta_{SD}}(\{s_t^{(i)}\}_{t=1}^{T}|z)$ the *state decoder* and $p_{\theta_{PD}}(\{a_t^{(i)}\}_{t=1}^{T}|z, \{s_t^{(i)}\}_{t=1}^{T})$ the *policy decoder*. Instead of having a separate policy decoder and control it to be consistent with the state decoder (as proposed by (Co-Reyes et al., 2018)), T-VAE models the policy and state decoder jointly and enables consistency inherently. Note that the state decoder does not depend on the policy and therefore during training, we do not need to interact with the environment nor conduct rollouts of the policy, making the learning process simpler and computationally relatively cheap.

For the state decoder, we assume Gaussian distribution on each $s_t^{(i)}$ where the variance is fixed and the mean $\hat{s}_t^{(i)}$ is computed recursively. At time $t <= T^{(i)}$, we concatenate $\hat{s}_t^{(i)}$ with the latent $z$ to form $[\hat{s}_t^{(i)}, z]$ which is fed into a LSTM cell, the output is then fed into a fully connected layer to produce $\hat{s}_{t+1}^{(i)}$. The fully connected layers guarantee that the dimensionality is preserved ($\hat{s}_t$ has the same dimension as $s_t$).

For the action decoder, we assume Gaussian distribution over $a_t$ for continuous actions and Multi-nomial/Bernoulli distributions for discrete actions. The variational parameter to be learned $\hat{a}_{t+1}^{(i)}$ is therefore the mean and the logits vector in the two cases respectively. Similarly as the state decoder, $\hat{a}_{t+1}^{(i)}$ is generated recursively from $[\hat{a}_t^{(i)}, \hat{s}_t^{(i)}, z]$ which is fed into a LSTM followed by a fully connected layer. Continuous actions are output at this stage, or an additional softmax/sigmoid activation function is applied to the output to generate discrete actions. If the action space consists of a mixture of continuous and discrete actions, we assume the actions are independent conditional on the states and latent variable, and the policy decoder can be factored as the product. An illustration of the entire model can be found in figure 1b with the bottom half representative of the decoder network.

### 3.2.3 VARIATIONAL BOUND

The marginal likelihood for each trajectory can be written as

$$\log p_\theta(\tau^{(i)}) = D_{KL}(q_\phi(z|\tau^{(i)})||p_\theta(z|\tau^{(i)})) + \mathcal{L}(\theta, \phi; \tau^{(i)}) \tag{2}$$

where $D_{KL}$ represents the KL divergence between the approximate posterior and the true posterior, and $\mathcal{L}(\theta, \phi; \tau^{(i)})$ is the variational lower bound of the marginal likelihood of $\tau^{(i)}$ which is decomposed into 3 terms: the $L2$ reconstruction loss for the state decoder, the $L2$ or cross entropy/sigmoid reconstruction loss for the policy decoder and a KL divergence between the posterior and prior distribution of the latent variable $z$. Formally:

$$\mathcal{L}(\theta, \phi; \tau^{(i)}) = \mathbf{E}_{q_\phi(z|\tau^{(i)})}[-\log q_\phi(z|\tau^{(i)}) + \log p_\theta(\tau^{(i)}, z)] \tag{3}$$

$$= \mathbf{E}_{q_\phi(z|\tau^{(i)})}[\log p_\theta(\tau^{(i)}|z)] - D_{KL}(q_\phi(z|\tau^{(i)})|p_\theta(z)) \tag{4}$$

$$= \mathbf{E}_{q_\phi(z|\tau^{(i)})}[\log p_\theta(\{s_t^{(i)}\}_{t=1}^T|z) + \log p_\theta(\{a_t^{(i)}\}_{t=1}^T|\{s_t^{(i)}\}_{t=1}^T, z)] - D_{KL}(q_\phi(z|\tau^{(i)})|p_\theta(z)) \tag{5}$$

As $\log p_\theta(\tau^{(i)}) \geq \mathcal{L}(\theta, \phi; \tau^{(i)})$, the encoder and decoder network parameters can then be optimised with stochastic gradient descent.

### 3.3 GENERATING TRAJECTORIES

After learning the latent representation, one can generate trajectories by: 1) sampling a $z$ in the latent space; 2) using the state decoder $p_{\theta_{SD}}(\{s_t^{(i)}\}_{t=1}^T|z)$ to decode the trajectory of states $\{s_t\}_{t=1}^T$; 3) applying the policy decoder $p_{\theta_{PD}}(\{a_t^{(i)}\}_{t=1}^T|z, \{s_t^{(i)}\}_{t=1}^T)$ to decode a sequence of $\{a_t\}_{t=1}^T$. In other words, all of the actions are predicted before the agent interacting with the environment. This may not be desired if the environment is noisy or the episode does not have a fixed length. Instead, one can use a rolling window to predict the trajectories for the next $n$ steps and refit the model with the new observation every $n$ steps, until the episode ends. We will discuss in more details the effect of the rolling window size $n$ later in section 4.3.

## 4 EXPERIMENT

### 4.1 2D NAVIGATION EXAMPLE

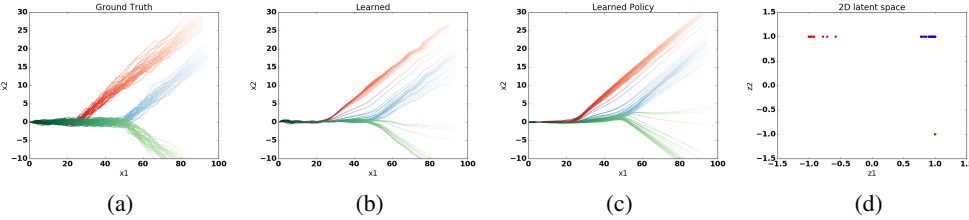

Figure 2: (a) Ground truth for test set; (b) reconstructed test set from state decoder; (c) reconstructed test set from policy decoder and (d) learned latent space for test set, each point in the latent space represents a trajectory.

We first apply our model to a 2D navigation example with 3 types of trajectories representative of players moving towards different goal locations. This experiment confirms our approach can detect and imitate multi-modal structure demonstrations, and learns a meaningful and consistent latent representation. Starting from $(0, 0)$, the state space consists of the 2D (continuous) coordinates and the action is the angle along which to move a fixed distance (=1). The time horizon is fixed to be 100.

In Figure 2, the ground truth trajectories are given in (a), and we reconstruct the trajectories through the state decoder and the policy decoder in (b) and (c) respectively. It can be seen that they are consistent with each other and represent the test set well. The latent embedding can be found in (d), where we can clearly identify 3 clusters corresponding to the 3 types of trajectories.

Figure 3 shows interpolations as we navigate through the latent space, i.e. we sample a 4 by 4 grid in the latent space, and generate trajectories using the state decoder and the policy decoder. We can see that the T-VAE shows consistent behaviour as we interpolate in the latent space. This confirms that our approach can detect and imitate latent structure, and that it learns a meaningful latent representation that captures the main dimensions of variation.

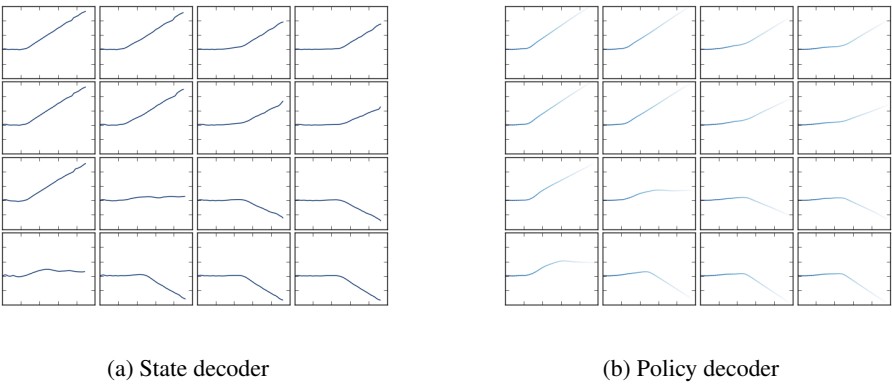

(a) State decoder            (b) Policy decoder

Figure 3: Intepolation of latent space for (a) state decoder; and (b) policy decoder. It can be observed that the top left corner, top right corner and bottom right corner behave like the red, blue and green type of trajectories respectively and the bottom left corner has a mixed behaviour.

## 4.2 2D CIRCLE EXAMPLE

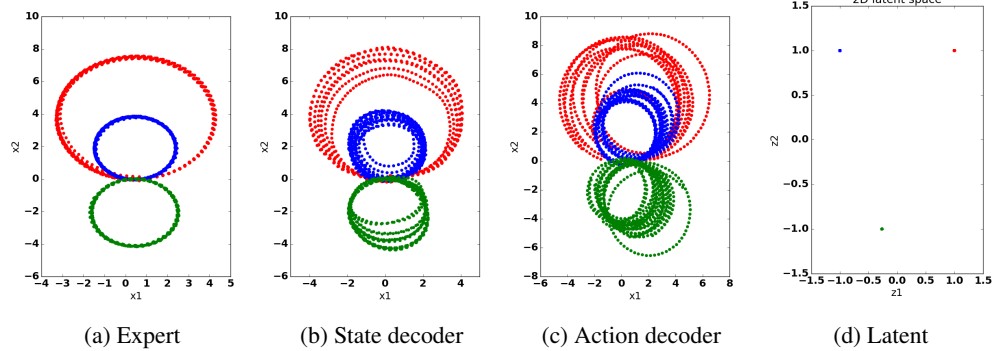

(a) Expert     (b) State decoder    (c) Action decoder    (d) Latent

Figure 4: (a):Ground truth of trajectories on the test set; (b): reconstructed trajectories with state decoder; (c) reconstructed trajectories with policy decoder; (d) 2D latent space.

We next apply our model to another 2D example, designed to replicate the experimental setting in Figure 1 of Li et al. (2017). There are three types of circles (in the figures these are coloured in red, blue and green) as shown in Figure 4a. The agent starts from $(0,0)$, the observation consists of the continuous 2D coordinates and the action is the relative angle towards which the agent moves. The reconstructed test set using state decoder and policy decoder, and visualisations of the 2D latent space can be found in Figure 4.

These results show that when the sequence length is not fixed (as in the previous example), T-VAE is still able to produce consistency between the state and policy decoders and learn latent features

that underpins different behaviours. Furthermore, as figure 1 in Li et al. (2017) already showed that both behaviour cloning and GAIL fail at this task whereas InfoGAIL and now T-VAE perform well, it seems that using a latent representation to capture long term dependency is crucial in this example.

### 4.3 ZOMBIE ATTACK SCENARIO

Finally, we evaluate our model on a simplified 2D Minecraft-like environment. This set of experiments show that T-VAE is able to capture long-term dependencies, model mixed action space, and the performance is improved when using a rolling window during prediction. In each episode, the agent needs to reach a goal. There is a zombie moving towards the agent and there are two types of demonstrated expert behaviour: the "attacking" behaviour where the agent moves to the zombie and attacks it before going to the goal, or the "avoiding" behaviour where the agent avoids the zombie and reaches the goal. The initial position of the agent and the goal are kept fixed whereas the initial position of the zombie is sampled uniformly at random. The observation space consists of the distance and angle to the goal and the zombie respectively, and there are two types of actions: 1) the angle along which the agent moves by a fixed step size (=0.5), and 2) a Bernoulli variable indicating whether to attack the zombie in a given timestep or not, which is very sparse and typically only equals to 1 once for the 'attacking' behaviour. Thus, this experiment setup exemplifies a mixed continuous-discrete action space. Episodes end when the agent reaches the goal or the number of time steps reaches the maximum number allowed, which is defined to be the maximum sequence length in the training set (30).

Figure 5 shows the ground truth and reconstruction of the two types of behaviours on the test set, and Figure 6 shows the learned latent space. We also provide animations: `https://youtu.be/fvcJbYnRND8` and 'avoiding' 'region'`https://youtu.be/DAruY-Dd9z8`. These show test time behaviour where we randomly sample from the posterior distribution of the latent variable $z$ in the latent space corresponding to the 'attacking' cluster.

To examine the diversity of the generated behaviour, we randomly select a latent $z$ in the 'attacking' and 'avoiding' clusters in Figure 6a and generate 1000 trajectories. The histogram for different statistics are displayed in Figure 7, where the top and bottom rows represent 'attacking' and 'avoiding' behaviour respectively. We can see a clear differentiation between these two different latent variables. Although the agent does not always succeed in killing the zombie, as shown in Figure 7b, the closest distances to the zombie (shown in Figure 7d) are almost all within the demonstrated range, meaning that the agent moves to the zombie but attacked at slightly different timing.

Results comparing with different rolling window length can be found in Figure 8. For the attacking agent, each episode is a success if the zombie is dead and the agent reaches the goal. For the avoiding agent, each episode is a success if the agent reaches the goal and is beyond the zombie's attacking range. It can be seen that for small rolling window lengths, the performance is worse for 'attacking' agent, since the model fails to capture long-term dependencies but provided a sufficient window length diverse behaviours can be imitated.

## 5 CONCLUSION

In this paper, we proposed a new method – *Trajectory Variational Autoencoder (T-VAE)* – for imitation learning that is designed to capture latent multi-modal structure in demonstrated behaviour. Our approach encodes trajectories of state-action pairs and learns latent representations with a VAE on the trajectory level.

T-VAE encourages consistency between the state and action decoders, helping avoid compound errors that are common in simpler behavioural cloning approaches to imitation learning. We demonstrate that this approach successfully avoids compound errors in several tasks that require long-term consistency and generalisation.

Our model is successful in generating diverse behaviours and learning a policy directly from a probabilistic model. It is simple to train and gives promising results in a range of tasks, including a zombie task that requires generalisation given a moving opponent as well as a mixed continuous-discrete action space.

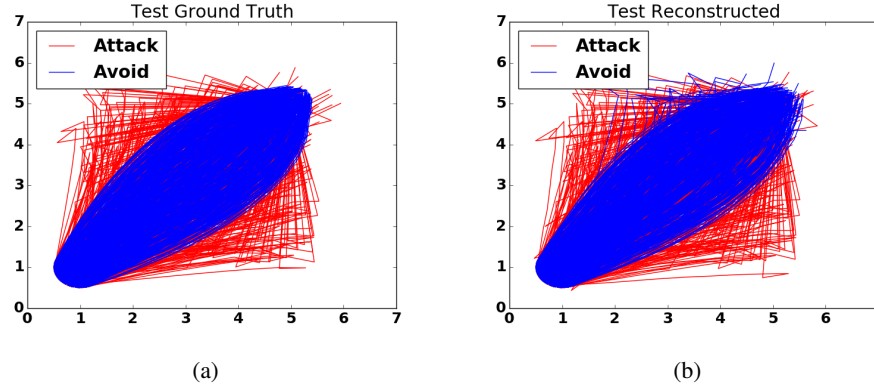

Figure 5: (a) Ground truth and (b) reconstruction (b) for the zombie attack scenario. The agent starts at $(0, 0)$, the goal is positioned at $(5, 5)$, and the zombie starts at a random location and moves towards the agent.

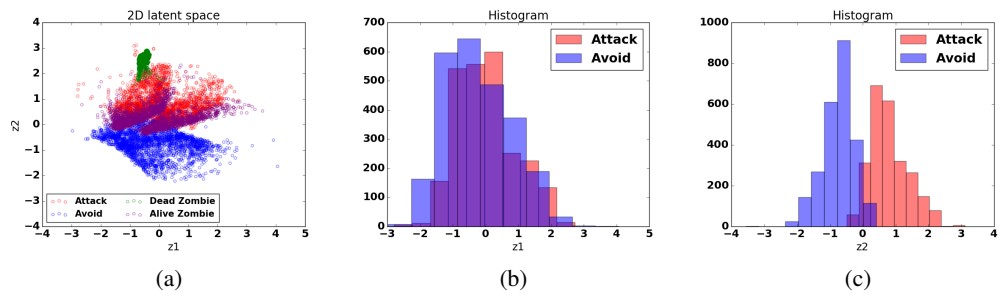

Figure 6: Latent representation of the zombie example which is clearly structured. The red and blue points represent the attacking or avoiding behaviour. We also encode partial trajectories before and after the zombie is dead for the 'attacking' agents, which are plotted in purple and green respectively.

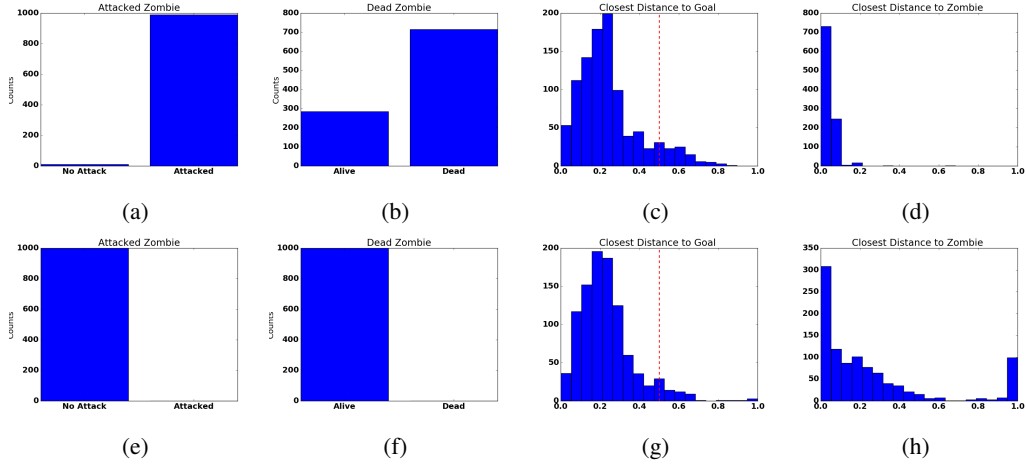

Figure 7: Top row and bottom row display the results of the trajectories generating from the 'attacking' and 'avoiding' cluster respectively. The first and second column show whether the agent attacks the zombie and whether the zombie is dead in the episode, the difference is sometimes agents attacks the zombie but are not in the attacking range so that the zombie does not die. The third and fourth column show the closest distance to the goal and the zombie in each episode. The agent reaches the goal when the distance to it is $< 0.5$ which is indicated by the red dash line (successful).

| | goals reached | | success rate | | dead zombie |
|---|---|---|---|---|---|
| n | attack | avoid | attack | avoid | attack |
| 5 | 84.30% | 90.50% | 0% | 53.00% | 0% |
| 10 | 89.50% | 90.70% | 20.50% | 52.00% | 28.40% |
| 15 | 91% | 99.50% | 62.90% | 38.70% | 71.50% |

Figure 8: Comparison of performance in the zombie attack scenario with varying window length.

A wide range of future work can be built upon ours. For example, bootstrapping reinforcement learning with these initial policies to improve beyond demonstrated behaviour provided an additional reward signal whilst aiming to maintain the diversity in behaviours.

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
