# OpenReview forum: "Trajectory VAE for multi-modal imitation"
_ICLR.cc/2019/Conference_

### Official Review · AnonReviewer3 · 2018-10-30

**Rating:** 4
**Confidence:** 4

**Review:**

The paper proposes an imitation learning model able to generate trajectories based on some expert trajectories. The assumption is that observed trajectories contain multi-modal (i.e. style) information that is not naturally captured by existing methods. The authors proposed a VAE based architecture that uses a prior distribution P(z) to simultaneously generate (state-action) pairs based on a LSTM decoder (actually, one LSTM for the states and one interleaved LSTM for the actions). This decoder is learned using a classical VAE auto-encoding loss, observed trajectories being encoder through a bi-LSTM. Experiments are made on three toy examples: a simple 2d Navigation case exhibiting 3 different 'styles', a 2D circle example with also 3 different styles, and a zombie attack scenario with two different styles. The results show that the model is able to capture different clusters of trajectories.

First of all, the paper does not propose a new model, but an instantiation of an existing model to a particular case. The main difference with SoTA is that the authors propose to both decode states and actions without using a simulator. The contribution of the paper is thus quite light. Moreover, it is unclear how the model can be used to get a policy corresponding to a particular mode. Can we use the learned decoders to generate actions on-the-fly in a real/simulated environment? Right now (section 3.3), actions are generated on generated states, but not on observed ones.  The paper has to clarify this point since just generating trajectories seems to be a little bit useless. In general Section 3.3 lacks of details (e.g the rolling window is also unclear). Also, the model could be described a little bit more in term of architecture, particularly on the critical point about how the two decoding LSTMs are interacting.

From the experimental point of view, the paper attacks very simple cases, without any comparison with state-of-the-art, and without almost any quantitative results. If Section 4.1 and 4.2 are useful to explore the ability of the model on simple cases, I would recommend the authors to merge these two sections in one smaller one, and then to focus on more realistic experiments. For example, it seems to me that the experimental setting proposed for example in [Li et al.] on driving styles could be interesting, and would allow a comparison with existing methods. Also the model proposed in [Co-Reyes et al.] could be an interesting comparison (at least, keeping the principle of this paper, without the hierarchical structure), particularly because this model is based on the use of a simulator while the proposed one is not. If a performance close to this baseline can be obtained with your model, it would be interesting for the community.

Right now, the experimental part and the too small contribution of the paper are not enough for acceptance. I would suggest the authors to:
* better describe their contribution i.e model architecture and how the model can be used to obtain a real policy
* use 'stronger' use cases for the experiments, and particularly existing use cases
* provide a deep quantitative and qualitative comparison with SoTA

Pro:
* simple method, no need of a simulator

Cons:
* not clear how to move from trajectory generation to a real policy
* small contribution
* too light experimental study without comparison with baselines and state of the art

---

> ### Author Response · Authors · 2018-11-26
> **Thank you for your constructive feedback.**
>
> We propose a new trajectory-level VAE which is different compared with previous work. The model is an alternative fully probabilistic model to capture state sequence dependencies, which is easy to train simply by gradient descent and has a promising performance on a range of problems. We agree that further experiments are needed as we mentioned in the comments for R1 and R2.  All the results we show for the rolling window case generate actions on-the-fly. The initial state is observed and a subsequent L actions are generated, after which the new observed state is fed into the model and so on. We will do more ablation studies to compare this with feeding the observed states directly into  the policy decoder during test, and clarify the experiment set up in section 3.3.

---

### Official Review · AnonReviewer2 · 2018-11-01
**Good but generic model, contribution limited**

**Rating:** 4
**Confidence:** 4

**Review:**


This paper proposes a VAE for modelling state-action sequences using a single latent variable rather than one per timestep. The authors empirically demonstrate that this model works on toy 2D examples and a simplified 2D Minecraft-like environment. Although I am unaware of other works that use a VAE in this setting, the model is still quite generic, thus requires further application to justify its significance. This paper is clear and well written.

The current contribution of this paper is limited, however it could be improved in a number of ways. The main component lacking from this paper is a meaningful comparison to other related works. Its unclear what the advantage of this model is over other models and so a thorough comparison to other sequence models would really help this paper. As mentioned in the conclusion, another direction for this work would be to bootstrap reinforcement learning. If this bootrapping was demonstrated then it would make this paper’s contribution stronger. Finally, another important direction for improvement for this paper would be to demonstrate its usefulness on more complex environments, instead of only 2D examples.

Pros:
- clear and well written
- model works on toy examples
Cons:
- lack of baseline comparisons
- lack of contributions

---

> ### Author Response · Authors · 2018-11-26
> **Thank you for your constructive feedback.**
>
> The main contribution of the paper is to introduce a consistent trajectory-level VAE which does not need simulation during training and serves as an alternative for capturing state sequence structure. We will strengthen our paper by experiments on the real MineCraft environment, quantitative comparison with SoTA algorithms and ablation studies, as well as the extension to bootstrapping reinforcement learning.

---

### Official Review · AnonReviewer1 · 2018-11-04

**Rating:** 4
**Confidence:** 4

**Review:**

This paper presents an approach to multi-modal imitation learning by using a variational auto-encoder to embed demonstrated trajectories into a structured latent space that captures the multi-modal structure. This is done through a stochastic neural network with a bi-directional LSTM and mean pooling architecture that predicts the mean and log-variance of the latent state. This is followed by a state and action/policy decoder (both LSTMs) that recursively generate trajectories from latent space samples. The entire model is trained by optimising the ELBO on a set of pre-specified expert demonstrations. At test time, samples are generated from the latent space and recursively decoded to generate state and action trajectories. The method is tested on three low-dimensional continuous control tasks and is able to learn structured latent spaces capturing the modes in the training data as well as generating good trajectory reconstructions.

Learning from multi-modal demonstration data is an important sub-area in imitation learning. As the paper pointed out, there has been a lot of recent work in this area. A lot of the ideas in this paper are similar to those proposed in prior work -- the network for embedding the trajectory is similar to the ones from Wang et al & Co-Reyes et al with the major difference being in the structure of the action decoder (and what inputs to encoder). Also, prior work has dealt with problems that are high-dimensional (Wang et al) and has shown results when operating directly on visual data (InfoGAIL). Comparatively, the results in this paper are on toy problems.

As there is no direct comparison to prior work provided in the paper, it is hard to quantify how much better the proposed approach is in comparison to prior work. For example, the "2D Circle Example" was taken from the InfoGAIL paper. It would have been good to use that as a baseline example to compare those two methods and highlight the advantages of the proposed approach -- did it require less data? fewer environment interactions? etc.

The results on the Zombie Attack Scenario seem poor. Specifically, in the avoid scenario, the approach seems to fail almost half the time. It would be good if the authors spend more time on this -- again, a comparison to prior work would establish some baselines and give us a good idea of the expected performance on this scenario. The videos show a single representative example for the "Attack" and "Avoid" scenarios. More examples including failures need to be included so that the distribution of results can be captured.

There is little in terms of generalisation or ablation studies in the paper. For example, in the Zombie Attack Scenario one could generate data with different zombie behaviours and measure performance on held out behaviours. Similarly, as an ablation, the authors could look at directly predicting actions instead of states & actions (states could be generated through a pre-trained dynamics model).

Figure 6. is hard to parse and could be explained better. No details are provided on the network architecture (number/size of the LSTM/fully connected layers), number of demonstrations used, training algorithm, hyper-parameters etc.

Few typos in the paper:
  Page 6 - between the animation links 'avoiding' 'region'
  Fig 7 caption - the zombie but are not in attacking range -> but the zombies are not in the attacking range,

Relevant citations that can be added:
1) Hausman, K., Chebotar, Y., Schaal, S., Sukhatme, G., & Lim, J. J. (2017). Multi-modal imitation learning from unstructured demonstrations using generative adversarial nets. In Advances in Neural Information Processing Systems (pp. 1235-1245).
2) Tamar, A., Rohanimanesh, K., Chow, Y., Vigorito, C., Goodrich, B., Kahane, M., & Pridmore, D. (2018). Imitation Learning from Visual Data with Multiple Intentions.

Overall, I find the paper to be incremental and lacking good experimental results and comparisons. The strengths of the paper are not clear and need to be explained and evaluated well. Substantial work is needed to significantly improve the paper before it can be accepted.

---

> ### Author Response · Authors · 2018-11-26
> **Thank you for your constructive feedback.**
>
> Our model differs from Wang et al in the sense that our VAE is on the trajectory level, which enables it to better identify the latent variable that differentiates different behaviors from the whole trajectory.  Co-Reyes et al  also uses a trajectory-level VAE, but our work differs from theirs in that our model is fully probabilistic and consistent. Therefore no extra penalty term is needed as in Co-Reyes et al to force the state decoder to be consistent with the action decoder.
>
> Thank you for the suggestions for the ablation studies. We will conduct a comprehensive analysis on the impact of the state decoder/policy decoder.
>
> We will fix the typos, add the references and clarify experiment setup in a revision.

---

### Meta-Review · Area_Chair1 · 2018-12-13
**Incremental solution, missing baselines**

**Confidence:** 4
**Recommendation:** Reject

**Metareview:**

The paper considers the problem of imitating multi-modal expert demonstrations using a variational auto-encoder to embed demonstrated trajectories into a structured latent space. The problem is important, and the paper is well written. The model is shown to work well on toy examples. However, as pointed out by the reviewers, given that multi-modal has been studied before, the approach should have been compared both in theory and in practice to existing methods and baselines (e.g., InfoGAIL). Furthermore, the technical contribution is somewhat limited as it using an existing model on a new application domain.